# Benchmarking Counterfactual Reasoning Abilities about Implicit Physical Properties

**Maitreya Patel, Tejas Gokhale, Chitta Baral, Yezhou Yang**
Arizona State University
{maitreya.patel, tgokhale, chitta, yz.yang}@asu.edu

## Abstract

Videos often capture objects, their motion, and the interactions between different objects. Although real-world objects have physical properties associated with them, many of these properties (such as mass and coefficient of friction) are not captured directly by the imaging pipeline. However, these properties can be estimated by utilizing cues from relative object motion and the dynamics introduced by collisions. In this paper, we introduce a new video question-answering task for reasoning about the implicit physical properties of objects in a scene, from videos. For this task, we introduce a dataset – CRIPP-VQA[1], which contains videos of objects in motion, annotated with hypothetical/counterfactual questions about the effect of actions (such as removing, adding, or replacing objects), questions about planning (choosing actions to perform in order to reach a particular goal), as well as descriptive questions about the visible properties of objects. We benchmark the performance of existing deep learning-based video question answering models on CRIPP-VQA (**C**ounterfactual **R**easoning about **I**mplicit **P**hysical **P**roperties). Our experiments reveal a surprising and significant performance gap in terms of answering questions about implicit properties (the focus of this paper) and explicit properties (the focus of prior work) of objects (as shown in Table 1).

## 1 Introduction

Visual grounding seeks to link visual signals such as images or videos, with natural language. Many tasks such as referring expressions [30], image captioning [26], image-text retrieval [27], and visual question answering [1] have been studied towards the goal of visual grounding. In the domain of video understanding, video question answering [13], video captioning [28], and text-based video retrieval [23] have been explored. Videos often contain objects, each having their own properties; for instance, objects belonging to certain categories, have shapes, sizes, and colors. These visible properties can be estimated by using computer vision algorithms for object recognition, detection, color recognition, shape estimation, etc. However, objects also have physical properties which in many cases are not captured by cameras. For example, cameras can capture the shape and color of an object, but not mass. Consider the frames in Figure 1 that contain objects with different shapes, textures, and colors, existing video question-answering datasets ask questions about these visible properties, but it is hard to reason about the masses of these objects or their coefficients of friction.

Collisions between objects and their change in velocity, however, do offer visual cues about mass and friction. When objects collide, their resulting velocities and direction of motion mainly depend upon their mass, according to fundamental Newtonian dynamics. By observing the change in velocities and directions, it is possible to reason about the relative physical properties of colliding objects.

---

[1]Project page: `https://maitreyapatel.com/CRIPP-VQA/`

36th Conference on Neural Information Processing Systems (NeurIPS 2022).

| Dataset | Video QA | Physical Reasoning | Visually Hidden Properties | Counterfactual Actions | | | Planning | Implicit reasoning |
|---|---|---|---|---|---|---|---|---|
| | | | | Add | Replace | Remove | | |
| MovieQA [25] | ✓ | - | - | - | - | - | - | - |
| TGIF-QA [18] | ✓ | - | - | - | - | - | - | - |
| TVQA/TVQA+ [17] | ✓ | - | - | - | - | - | - | - |
| AGQA [10] | ✓ | - | - | - | - | - | - | - |
| CoPhy [3] | - | ✓ | ✓ | - | - | - | - | ✓ |
| CLEVR_HYP [24] | - | - | - | ✓ | ✓ | ✓ | - | - |
| IntPhys [22] | ✓ | ✓ | - | - | - | - | ✓ | - |
| ESPRIT [21] | ✓ | ✓ | - | - | - | - | ✓ | - |
| CATER [9] | ✓ | - | - | - | - | - | - | - |
| CRAFT [2] | ✓ | ✓ | - | - | - | ✓ | - | - |
| CLEVRER [29] | ✓ | ✓ | - | - | - | ✓ | - | - |
| ComPhy [5] | ✓ | ✓ | ✓ | - | - | - | - | - |
| **CRIPP-VQA (ours)** | ✓ | ✓ | ✓ | ✓ | ✓ | ✓ | ✓ | ✓ |

Table 1: A comparison of CRIPP-VQA with prior work on video question answering, in terms of different aspects of visual reasoning that are tested.

In many cases, When humans watch objects in motion and under collision, we do not accurately know the masses, friction, or other properties of objects. Yet, when we interact with these objects, for example in a game of billiards, we can reason about the effect of actions such as hitting one ball with another, removing an object, replacing an object with a different one, or adding an object to the scene. In this paper, we consider the task of reasoning about such implicit properties of objects, via the use of language, without having annotations for the true values of mass and friction of objects. We propose a video question answering dataset called CRIPP-VQA, short for **C**ounterfactual **R**easoning about **I**mplicit **P**hysical **P**roperties [20]. CRIPP-VQA contains videos annotated with question-answer pairs. Each video contains several objects with at least one object in motion. The object in motion causes collisions which changes the spatial configuration of the scene. The consequences of the collision are directly impacted by the physical properties of objects. CRIPP-VQA asks questions about these consequences.

As shown in Figure 1, questions in the dataset require understanding the current configuration as well as counterfactual situations, i.e. the effect of actions such as removing, adding, and replacing objects. The dataset also contains questions that require the ability to plan in order to achieve certain configurations, for example producing or avoiding particular collisions. It is important to note that both tasks can not be performed without an understanding of the relative mass. For example, replace action can lead to a change in mass inside the reference video, which can drastically change the consequences (i.e., a set of collisions).

We benchmark existing state-of-the-art deep learning-based video question-answering models on the new CRIPP-VQA dataset. Our key finding is that compared to performance on questions about visible properties ("descriptive" questions), the performance on counterfactual and planning questions is significantly low. This reveals a large gap in understanding the physical properties of objects from video and language supervision. Detailed analysis shows that the models can predict the first collision on counterfactual questions with high accuracy compared to the subsequent collisions. Models struggle at answering questions about the effect of *"replace"* action. In the case of the *"add"* action, models can predict which collisions won't happen, but fail to predict the collisions that indeed happen.

Recently, many neuro-symbolic approaches are proposed for CLEVRER-like settings. For example, IEP [15], NS-DR+ [19], are CPL [5] proposed for physical reasoning. However, the goal of our study is to evaluate whether systems can learn the implicit relationship from counterfactual tasks. However, symbolic approaches either require providing this information explicitly or learning through the physics engine, which is not feasible for real-life situations. Therefore, in this study, we put forward this important and difficult problem of learning visually hidden properties in an implicit manner to motivate the work in the neuro-symbolic domain.

We summarize our contributions below:

1. We introduce a new benchmark on counterfactual reasoning, CRIPP-VQA, for video question answering which requires implicit reasoning about the physical properties of objects in videos.

2. CRIPP-VQA contains questions about the effect of actions such as removing, replacing, and adding objects, which have not been considered in prior work on video QA.
3. We specifically show that previous state-of-the-art methods do not perform as expected and a lot of work needs to be done to achieve human-level performance.

# 2 The CRIPP-VQA Dataset

CRIPP-VQA, short for Counterfactual Reasoning about Implicit Physical Properties via Video Question Answering, focuses on understanding the consequences of different hypothetical actions (i.e., remove, replace, and add) in the presence of mass and friction as visually hidden properties.

## 2.1 Simulation Setup

**Objects and States.** Each object in the CRIPP dataset has four visible properties: shape (cube or sphere), color (olive, purple, and teal), texture (aluminum and cardboard), and state (stationary, in motion, and under collision). Each object also has two invisible properties: mass and coefficient of friction. Three actions can be performed on each object – "remove", "replace", and "add".

In this work, we focus on mass and friction as intrinsic physical properties of objects. Each unique {SHAPE, COLOR, TEXTURE} combination is pre-assigned a mass value (in whole dataset) that is either 2 or 14; for instance, all teal aluminum cubes have mass 2. Note that these values are not provided as input to the VQA model and need to be inferred in order to perform counterfactual and planning tasks. In the CRIPP-VQA dataset, the coefficient of friction for all objects with the surface is identical and non-zero (i.e., 0.25).

**Video creation.** We render videos using *TDW* [8]. Firstly, in each instance, we initialize the video with either 5 or 6 randomly chosen objects, out of which a single object will be initialized with a fixed velocity such that it will collide with other objects. Here, we keep a constant initial velocity so that the only way to infer mass is through the impact of subsequent collisions. Each video is 5 seconds long, with a frame rate of $25 fps$. We provide annotation and metadata for each video which contains object locations, velocities, orientation, and collision info at each frame. These annotations are further used to generate the different types of question-answer pairs.

## 2.2 Question and Answer Generation

CRIPP dataset focuses on three categories of tasks: 1) Descriptive, 2) Counterfactual, and 3) Planning.

**Descriptive:** These questions involve understanding the visual properties of the scene, including:

1. Counting the number of objects having a certain combination of visually seen properties,
2. Yes/No questions requiring object recognition
3. Finding the relationship between two objects under collision
4. Counting the number of collisions
5. Finding the maximum/minimum occurring object properties.

We do not include questions that require reasoning over mass, to avoid the introduction of spurious correlation which may influence counterfactual and planning-based questions.

**Counterfactual.** These questions focus on action-based reasoning (i.e., remove, replace, and add). We generate a hypothetical situation based on one of these actions, and the task is to predict which collisions may or may not happen if we perform the action on an object. **Remove** action focuses on a counterfactual scenario where a certain object is removed from the original video. **Replace** action focuses on a counterfactual scenario where one object is replaced with a different object. Replace action does not only change the object but it may also lead to a change in the hidden property. **Add** action-based questions focus on evaluating the system's understanding of spatial relationship along with the hidden property, where we create a new hypothetical condition by placing a new object to the $left/right/front/back$ at a fixed distance from the reference object.

| Model | Descriptive | Remove | | Replace | | Add | | Planning |
|---|---|---|---|---|---|---|---|---|
| | | PQ | PO | PQ | PO | PQ | PO | |
| Frequency | 8.21 | 0.00 | 50.18 | 0.00 | 50.00 | 0.00 | 50.00 | 3.49 |
| Random | 8.51 | 7.21 | 49.58 | 3.34 | 49.40 | 9.39 | 50.04 | 7.39 |
| Blind-BERT | 53.82 | 20.18 | 54.67 | 17.57 | 50.45 | 15.86 | 51.55 | 8.11 |
| MAC [12] | 48.72 | 16.41 | 50.68 | 17.31 | 50.21 | 16.29 | 49.83 | 6.26 |
| HCRN [16] | 64.98 | 27.20 | 59.04 | 19.87 | 55.97 | 20.49 | 56.06 | 21.38 |
| Aloe* | 68.94 | 31.10 | 62.90 | 9.91 | 52.10 | 18.13 | 56.55 | 31.76 |
| Aloe*+BERT | 71.04 | 33.64 | 65.46 | 22.07 | 56.76 | 39.71 | 67.43 | 32.61 |

Table 2: Results on the test set showing performance of models evaluated in terms of per-question (PQ) accuracy and per-option (PO) accuracy. For counterfactual questions are multiple choice based, therefore per-question and per-option accuracies are shown.

**Planning.** CRIPP also contains planning-based questions, where the task is to perform an action on objects within the given video to either $make/stop$ collisions between two given objects. Here, the system needs to predict which action needs to be performed and on which object, to achieve the goal.

## 2.3 Dataset Statistics

CRIPP contains 4000, 500, and 500 videos for training, validation, and testing, respectively. Furthermore, it has about 2000 videos focused on evaluation for physical out-of-distribution scenarios. CRIPP training dataset has about 41761 descriptive questions, 41761 counterfactual questions (9603, 5142, and 27016 questions for remove, replace, and add actions, respectively), and 10440 planning-based questions.

# 3 Experiments

## 3.1 Problem Statement

Given an input video ($v$), and a question ($q$) the task is to predict the answer ($a$). Each video $v$ contains the $m$ number of objects randomly selected from the set $O = \{o_1, o_2, ..., o_n\}$. Here, object $o_i$ has several associated properties (i.e., $o_i = (m_i, c_i, s_i, t_i, l_i, v_i)$), where color ($c_i$), shape ($s_i$), texture ($t_i$), location ($l_i$), and velocity ($v_i$) are visually observable properties alongside with mass ($m_i$) as hidden property. More formally, we need to learn the probability density function $F$ such that we maximize the $F(a|v, q)$.

**Evaluation Metrics.** To evaluate the models, we use two accuracy metrics – per-option (PO) and per-question (PQ) accuracy. Here, each counterfactual questions have multiple choices describing possible set of collisions. Therefore, per-option accuracy refers to the choice-wise performance and per-question accuracy considers whether all choices are correctly predicted or not. Moreover, each planning task involves performing an action over objects within a video. Because of that to achieve the given goal, there can be multiple solutions. Therefore, we use *TDW* to re-simulate the models' predictions on the original video to check whether the given planning goal is achieved or not. Therefore, for the planning-based question, we do an iterative simulation-based evaluation.

## 3.2 Benchmark model details

We consider three different deep learning-based state-of-the-art models for the video question-answering task: MAC [12], Hierarchical Conditional Relation Network (HCRN) [16], and 3) Attention over learned embeddings (Aloe) [7]. **MAC** is designed for compositional VQA. We modify it by performing channel-wise feature concatenation of each frame, where the channel will contain temporal information instead of spatial information allowing MAC to adapt to the video inputs. **HCRN** uses a hierarchical strategy to learn the relation between the visual and textual data. **Aloe** is one of the best-performing models on the CLEVRER [29] benchmark. It is a transformer-based model, designed for object trajectory-based complex reasoning over synthetic datasets. Aloe uses MONet [4] for obtaining object features by performing an unsupervised decomposition of each frame into observed objects. Aloe takes these frame-wise object features to predict the answers to the input question, using the *[CLS]* token and self-supervised training strategy.

**Aloe\* (Modified Aloe)**   We find that the MONet module used in Aloe is very unstable and fails to produce reliable frame-wise features on complex visuals from CRIPP-VQA. Because of this Aloe model fails measurably on CRIPP-VQA, exhibiting close-to-random performance. Therefore, we propose additional modifications to Aloe to make it more widely applicable beyond prior datasets that are built using the CLEVR [14] rendering pipeline. First, we replace MONet with Mask-RCNN [11] to perform instance segmentation and then train an auto-encoder to compress the mask-based object-specific features to make it compatible with Aloe. Second, instead of learning the word embedding from the scratch, we further propose to use BERT-based word embeddings as input to the Aloe, which leads to faster and stable convergence.

In addition to these baselines, we also consider a *"random"* baseline which randomly selects one answer from a possible set of answers, and a *"frequent"* baseline which always predicts the most frequent label. To analyze textual biases, we use a text-only QA model and denote it by *"Blind-BERT"*. Blind-BERT is a pre-trained language model (BERT [6]) which takes only questions as input to predict the answer, and ignores the visual input.

### 3.3   Results

Table 2 summarizes the performance comparisons of our baselines on the CRIPP-VQA test set. On **Descriptive** questions, the *"random"* and *"frequent"* baselines achieve around only $8\%$ accuracy, while Blind-BERT gets 53.82% which suggests the existence of language bias associated with correlations between question types and most likely answers for each. Surprisingly, MAC achieves only 48.72% which is lower than Blind-BERT. This implies that the video feature representations learned by MAC hurt performance compared to text-only features. HCRN and both Aloe variants (Aloe\* and Aloe\*+BERT) improve performance indicating that visual features are crucial for descriptive questions. Aloe\*+BERT is the best performing model which implies that our modification with BERT features helps performance.

**Counterfactual & Planning** tasks involve a total of three types of actions. The performance of MAC is again close to Blind-BERT. HCRN performs slightly better than Blind-BERT. This shows that even though visual features in HCRN are better than the MAC but it is not sufficient enough to do such complex reasoning. Aloe\*+BERT achieves much better results only in terms of remove and add actions. However, Aloe\*+BERT is close to random for questions with the *"replace"* action as it directly involves the change in physical properties (i.e., mass and shape) of an existing object within the given scenario. This implies that Aloe\*+BERT is able to do spatial reasoning to some extent, but is not good at reasoning about changes in physical properties. While it can also be seen that Aloe\*+BERT outperforms the Aloe\* across the actions, this implies that BERT-based embedding helps the model to learn the relation between the objects and action.

**Human evaluations.** To learn more about the upper bound of CRIPP-VQA dataset and what to expect from the different proposed systems, we perform human evaluations. There were total 6 people participated as volunteers. All were given 5 video and corresponding QA pairs to get habituated with the environment. Then we asked them to answer total 30 questions. Results shows that Human evaluations achieved 90.00%, 78.89%, and 58.87% on descriptive, counterfactual (per-option accuracy), and planning tasks, respectively.

## 4   Conclusion and Future Work

In this work, we present a new benchmark: CRIPP-VQA, for counterfactual/hypothetical reasoning about the implicit physical properties of objects in a scene. We propose novel counterfactual reasoning and planning tasks, over three hypothetical actions (i.e., remove, replace, and add). We evaluate state-of-the-art models on this benchmark and observe a significant performance gap between descriptive questions about visible properties and counterfactual and planning questions about implicit properties.

Based on the prior work, we hypothesize that neuro-symbolic approaches might lead to better performance on CRIPP-VQA. Especially, when we can incorporate Newtonian dynamics to design physics-aware neuro-symbolic methods. This result is positioned as a challenge for the neuro-symbolic research community to design methods which can learn symbolic knowledge without using explicit modules.

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

| Hyper-parameter | Value |
| --- | --- |
| # of layers | 28 |
| # of attention heads | 128 |
| embedding size | 768 |
| visual feature size | 512 |
| text embedding size | 768 |
| Batch Size for descriptive | 96 |
| Batch Size for Counterfactual | 32 |
| Batch Size for Planning | 16 |
| Learning rate | 0.00005 |
| Optimizer | RAdam |

Table 3: Aloe+BERT architecture and hyper-parameter details.

# A   Appendix

## A.1   Training details

We follow the standard training guidelines provided by the authors of each paper. We train all systems on Quadro RTX 8000 GPUs. We train each model with a maximum of 200 epochs. And select the best model based on average performance accuracy. We follow the below instructions to support each model which are MAC, HCRN, Aloe, and Aloe+BERT. Moreover, for planning based task, we add extra four classifier heads on top of all models which predicts: 1) the type of the action, 2) an object on which action needs to be performed, 3) an object which needs to be added through replace or add action, and 4) relative direction of the object if we are adding a new object.

**MAC:** We modify the public implementation of MAC from `https://github.com/rosinality/mac-network-pytorch` to adapt the video frames as input. We first resize the each 125 frames leading $(125, 3, 224, 224)$ video dimension. Later, we use ResNet101 to extract the features $(125, 512, 14, 14)$. After taking the channel-wise mean of features, we get the final video representation of $(125, 14, 14)$ dimension matrix supportable for the rest of the pipeline. We also do the necessary changes described for the planning task as well.

**HCRN:** As HCRN is the VideoQA model and official implementation is available at: `https://github.com/thaolmk54/hcrn-videoqa`, we use the source code as it is. Except we do important changes to do planning tasks.

**Aloe/Aloe+BERT:** We first reproduce the Aloe on PyTorch based on the architecture details from the research paper by Ding et. al. [7] and their public available demo at `https://github.com/deepmind/deepmind-research/tree/master/object_attention_for_reasoning`. Moreover, we use the code base from transformers[2] library (as it is well-tested and used across the industry and academia) and modify it to support the VideoQA in the same way as Aloe does. Our initial experiments on CLEVRER showed that Aloe cannot reproduce the results on CLEVRER with the specified set of architecture details and hyper-parameters from the original paper. Therefore, we do extensive experiments on Aloe architecture and hyper-parameter to reproduce similar results. After achieving a similar performance from the paper, we use this new reproducible Aloe architecture in our experiments. Table (3) shows the hyper-parameter details to reproduce the results. Moreover, the Aloe source code from our experiments is available at `https://maitreyapatel.com/CRIPP-VQA/`

---

[2]`https://github.com/huggingface/transformers`

## Time

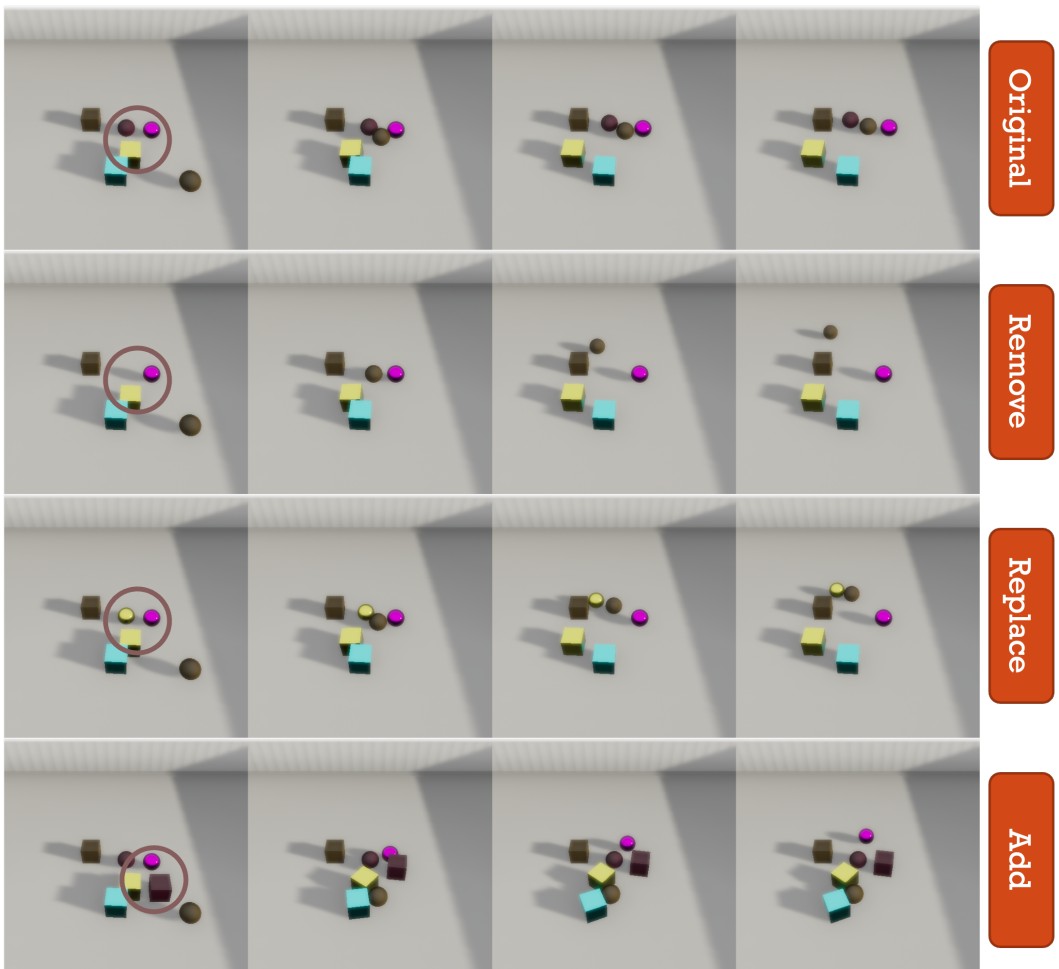

**Counterfactual Questions:**
1. What will happen if we remove the purple cardboard sphere?
2. What will happen if we replace the olive cardboard sphere with an olive aluminum sphere?
3. What will happen if we add a purple cardboard cube to the right of an olive aluminum cube?

**Planning Questions:**
1. Make the collision between the olive cardboard sphere and the teal aluminum cube.
2. Stop the collision between the olive cardboard sphere and the purple aluminum sphere.

Figure 1: The CRIPP-VQA dataset contains questions about the future effect of actions (such as removing, adding, or replacing objects) as well as planning-based questions. A few frames of a video are shown above, with the red highlighted area depicting the objects on which actions are performed.

