# OpenReview forum: "Benchmarking Counterfactual Reasoning Abilities about Implicit Physical Properties"
_NeurIPS.cc/2022/Workshop/nCSI — nCSI WS @ NeurIPS 2022 Poster_

### Official Review · Reviewer_yf1U · 2022-10-14
**A new VQA benchmark yields hard to interpret results**

**Rating:** 2
**Confidence:** 2

**Review:**

The paper introduces a new benchmark dataset containing synthetic videos of colliding objects, and generated questions about them. It is designed to explore the ability of VQA models to infer latent physical properties, such as object mass. This ability is challenged through counterfactual ("what would happen if an object is added/replaced/removed?") and planning ("what action should be taken on an object to produce/prevent a certain collision") questions. A number of recent VQA models, and modifications thereof, are evaluated on the new benchmark.

Pros:
 - The paper investigates an interesting question relevant to the workshop, namely how VQA models may infer latent object properties for better counterfactual reasoning.
 - The dataset design is generally sensible.
 - The Aloe model is adapted to the new dataset to ensure a fair comparison.

Cons:
 - While the stated purpose of the new dataset is to investigate the relevance of latent object properties, it remains unclear to what degree the observed results are a consequence of the latent mass differences. Performance disparities could also result from the new visual setting (as was observed for MONet), or the fact that counterfactual questions are generally harder than descriptive ones. It would have been helpful to either directly measure the models' ability to predict mass through a descriptive question, or to compare results with a dataset variant in which masses are identical for all objects.
 - The paper somewhat overlaps with other recent work on VQA benchmarks, such as ComPhy. The authors argue that their dataset differentiates itself by featuring both hidden properties and counterfactual add/replace/remove questions, but, for the reasons above, I feel like this doesn't necessarily make analyzing the results any easier.
 - While the paper is generally clear, the text, especially section 3, contains a number of grammatical errors, e.g. "[...] whether all options of correctly predicted or not". I was also left with a number of questions that would be good to clarify (see below).

Despite these weaknesses somewhat limiting the paper's significance, I still think it will make a reasonable contribution to a question that is very relevant for the nCSI workshop.

Questions:
 - Are the mass values of certain object types (e.g. teal aluminum cube) consistent throughout the dataset, or resampled for each scene? I assume the latter, as otherwise models would not need to infer it at test time.
 - What exactly is meant by the term "option"?
 - Are the human performance scores per option or per question?
 - When a counterfactual question introduces a new object, does the model always have a chance to infer the latent mass, or is it possible that previously unseen object identities are introduced?

---

### Official Review · Reviewer_zGNf · 2022-10-15
**Video QA dataset for learning implicit physical properties of objects to do counterfactual and planning tasks**

**Rating:** 2
**Confidence:** 2

**Review:**

Overall I find the paper good. They have multiple tasks to learn about the implicit and explicit object properties. They show the gap that exists currently between learning to do descriptive tasks from visual features and to do counterfactual or planning tasks using implicit properties. They have done a great work to create scenarios to learn these properties. However, I believe, the range of values for these implicit features should have been more which could have made this dataset even more useful.

# Pros
- it's a nice benchmarking dataset with a varied range of scenarios for counterfactual reasoning.
- the collection of tasks is quite fascinating. the way they re-simulate and do iterative performance evaluation for planning tasks was quite good.
- one very interesting is that the only way to estimate mass is through subsequent collisions since object are initialised with fixed velocity. this way they make sure the model doesn't cheat and learn about mass through momentums.


# Cons
- it was a bit hard to differentiate colours in the video. that could have been made better. I believe this might be a problem during model training since most same-shaped objects looked too dark and of similar colour.
- there are only two mass values for the objects. 2 and 14. I understand this might have been done to avoid the exponential increase in the possible state spaces. However, having more values would have been nicer.
- similar opinion for friction coefficients

---

### Meta-Review · Area_Chair_aKJN · 2022-10-17

**Recommendation:** 2
**Confidence:** 3

**Metareview:**

The paper aims at benchmarking deep video question answering models in particular w.r.t counterfactual reasoning, i.e., whether they may infer latent object properties for better counterfactual reasoning. To this end, it introduces a dataset CRIPP-VQA. The empirical results show a gap in terms of answering questions about implicit and explicit properties. Both reviewers agree that this is solid work and should be accepted: The dataset design makes sense and the evaluation is fair. I fully agree.

---

### Decision · Program_Chairs · 2022-10-20

Accept (Poster)